# Valorisation of Forest and Agriculture Residual Biomass—The Application of Life Cycle Assessment to Analyse Composting, Mulching, and Energetic Valorisation Strategies

Isabel Brás [1,2,*], Elisabete Silva [1,2,3], Roberta Raimondo [4], Raffaella Saetta [4], Vincenzo Mignano [4], Massimiliano Fabbricino [4] and José Ferreira [5]

1    Department of Environment, School of Technology and Management, Polytechnic University of Viseu, 3504-510 Viseu, Portugal; beta@estgv.ipv.pt
2    CISeD—Research Centre in Digital Services, Polytechnic University of Viseu, 3504-510 Viseu, Portugal
3    LEPABE—Laboratory for Process Engineering, Environment, Biotechnology and Energy, Faculty of Engineering of the University of Porto, 4200-465 Porto, Portugal
4    Civil Engineering, Construction and Environment Department, University of Naples Federico II, 80125 Nápoles, Italy; rob.raimondo@outlook.com (R.R.); raffa97107@gmail.com (R.S.); vinc.mignano@studenti.unina.it (V.M.); massimiliano.fabbricino@unina.it (M.F.)
5    CERNAS-IPV Research Centre and ESTGV, Polytechnic University of Viseu, 3504-510 Viseu, Portugal; jvf@estgv.ipv.pt
*    Correspondence: ipbras@estgv.ipv.pt; Tel.: +351-232480500

**Abstract:** The treatment of agricultural and forest residues (AFRs) has become an important issue nowadays, both to avoid improper management and for their enhancement. In the study area of Viseu (Portugal), the AFRs are taken to a Residual Biomass Collection Centre. These are valorised in a Biomass Power Plant to produce electricity. Two further processes could be implemented to valorise this biomass: mulching and composting. This study aims to understand the best strategy to enhance this type of biomass residual considering their environmental performance. The Life Cycle Assessment (LCA) was applied considering a cradle-to-grave approach. Different processes of all the technologies were analysed, and the data collected enabled a comparison of 11 environmental impact categories. The results show that composting is the best alternative, except for global warming and ozone layer depletion, and energy valorisation has the greatest impact on five of the considered categories. In the three processes, impacts are mainly associated with the production and shredding phases of the residual biomass, rather than the transportation stages, due to the short distances covered. In all cases, the value of the final product generates market consequences in terms of electricity and fertiliser production. In line with the aim of the study, its outcomes may provide scientific support to local decision makers in defining best practices in the management of the AFRs.

**Keywords:** agricultural and forest residues; biomass valorisation; composting; mulching; energetic valorisation; environmental impact; life cycle assessment

## 1. Introduction

In recent years, the risk of forest fires has increased significantly in the municipality of Viseu, located in northern Portugal. Their occurrence is not only a consequence of climate change but also of the improper maintenance of forests and the mismanagement of forest residues [1]. This phenomenon has a significant economic, environmental, and social impact. Even though it is mandatory by law [2] to collect residual materials in forests or pastures, private landowners often disregard this practice due to the costs of operation and disposal. Therefore, in most cases, due to the difficulties in managing waste on-site, the final treatment of this forest waste is burning in the same area where it is produced. This practice is the main cause of rural fires in Portugal and worldwide [3–5]. In order to comply with the above-mentioned decree-law, the Municipality of Viseu has a network of Residual

Biomass Collection Centres (RBCCs) through which it aims to enhance the collection of agricultural and forest residues throughout the region. This is not only to reduce the risk of rural fires but, above all, to valorise this important residual bio-resource. As waste, the following management hierarchy policy should be followed: prevention, preparation for re-use, recycling, valorisation such as energy recovery and, finally, disposal. However, it may be necessary for specific waste streams to diverge from this hierarchy when this is justified by the life cycle approach in relation to the global impacts of waste generation and management [6]. It is necessary to rethink these wastes from a circular economy perspective, in which all the bio-resources they contain must be valorised. These actions can take place in different ways; indeed, the different values, not only environmental, of the specific recovered element must be taken into account [7]. Among the proper management practices for agricultural and forest residues (AFRs), currently the most widely used is their transfer to dedicated collection centres, RBCCs, to be transported to a biomass power plant for energetic valorisation. This study seeks to explore some alternatives to this practice, evaluating, when possible, different techniques; in particular, techniques that involve the deposition of organic material on the soil, compost, or mulch to enrich it with bio-chemical nutrients and improve its physical and chemical characteristics. Activities that aim to recover material from waste before it is used to produce energy are also encouraged by the sustainable development goals set out in the European 2030 agenda action programme [8]. Therefore, it is intended to investigate how the utilisation of forest and agricultural residues would maximise the value of these residual biomass, offering an alternative to energy valorisation [9]. Several processes may be applied to recover carbon and other nutrients from the residual biomass, namely, biomass gasification [10]; pyrolysis [11]; torrefaction [12]; dark fermentation [13]; or using biomass residual in biorefinery applications to produce bioethanol, biodiesel, biohydrogen, biogas, organic acids, biomaterials, bio-oil, and various pharmaceutical and nutraceuticals compounds [14,15]. The most efficient way to examine the entire environmental impact of all energy and material flows, whether input or output, over the course of a product's life cycle is through a life cycle assessment (LCA) [16]. LCA is the internationally recognised standard approach for assessing the environmental impact of a product, by which environmental effects can be calculated holistically, considering both the material and energy resources consumed and the emissions generated [17]. With the use of this technique, it is possible to quantify the current environmental areas most affected by impact and, at the same time, identify areas that could be improved to reduce potential negative environmental effects in the future [18]. Particularly, the purpose of this study is to compare three different techniques for the valorisation of these bio-resources considering their local application: composting, mulching, and energy valorisation. These techniques are the easiest to implement in the socio-economic system of the study area under consideration. The first two do not need special technologies and can be carried out in the same place where the AFRs are produced. In addition, the bio-resources obtained can also be used on-site, thus reducing the economic and environmental costs associated with transport. The third technology, instead, is that one currently used to valorise this type of residue.

The composting technique allows the production of compost, a product with a high content of stabilised organic substances, in particular humic acids, capable of improving the physical and biochemical characteristics of the soil. Its distribution on the soil also counteracts erosion phenomena and allows a progressive accumulation of carbon in the soil. It is a carbon-capture technique used to mitigate climate change. On the other hand, it increases the amount of nutrients in the soil, improving its fertility. Hence, it can be used to supplement or replace chemical fertilisation, the reduction of which can have important environmental and economic implications [19]. Mulching of lignocellulosic materials can provide benefits such as controlling weeds, reducing evaporation, and moderating soil temperature, in addition to revitalising soil fauna and restoring natural forests on degraded soils [20]. Finally, energy valorisation allows the recovery of electricity in an existing plant located within the study area through a traditional waste-to-energy process.

The most recent studies on the subject show how efficient use of biomass residues from agricultural and forestry activities significantly reduces environmental impacts without endangering global forests rather than protecting them from future risks. As already said, the LCA results identify the most significant causes of environmental degradation, allowing efforts to focus on reducing these impacts [21]; at the same time, the results demonstrate how some impact categories are increased when using specific techniques.

## 2. Material and Methods

The comparative life cycle assessment of the mulching, biological, and thermal valorisation processes has been carried out according to ISO 14040 [22], ISO 14044 [23], and the International Reference Life Cycle Data System (ILCD). The LCA procedure is structured in four steps [24]: goal and scope definition, Life Cycle Inventory (LCI), Life Cycle Impact Assessment (LCIA), and interpretation and presentation of results. The study was carried out with the software SimaPro 9.3.0.3 PhD, in which the different processes are simulated, and results are obtained. The Impact Assessment method chosen to compare the obtained results was the CML-IA baseline V3.07/World 2000.

### 2.1. Goal and Scope Definition

The goal of the study is to compare the environmental burden related to three techniques for AFR valorisation—energy valorisation, composting, and mulch production—through a cradle-to-grave approach. AFRs are collected at the RBCC of Bodiosa, located north of the municipality of Viseu, whose location is important to define the distance between the centroids of the study area and the collection point.

#### 2.1.1. System Boundaries

To reach the goal of the study, the system boundaries were defined for the three options, beginning with the production of the AFRs and subsequent transportation to the RBCC, where the shredding is performed. The mulching process merely consists of the transportation of the shredded biomass to agricultural fields and forestry for manual distribution and use (Figure 1). For composting, after the shredding of the AFRs, the process includes the following steps: construction of cone-shaped piles with AFRs, production of mature compost after 120 days, compost transportation to agricultural fields, and manual distribution on the agricultural fields, where it is used as an organic soil conditioner (Figure 2). Only the concentration of macronutrients (N, P, and K) in the produced compost is considered in the study. It disregards considerations concerning compost quality and its actual compliance with legislative values. This aspect is a limitation of the study. For both composting and mulching, the boundaries of the system were extended to include the production of chemical fertiliser replaced by compost. The mulch is assessed based on their nutrient content, and the destination of the nutrients that are not absorbed by plants was also considered in the study, since they represent an environmental burden. The influence of the obtained products on the carbon cycle downstream of their distribution is excluded from the system boundaries due to the complexity of the phenomena. For the energetic valorisation, the transportation to the Biomass Power Plant, as well as the combustion process, the feed water, and ashes treatment, were considered. The system boundary of the electricity production from biomass was extended to include the avoided electricity (high voltage) produced from the Portuguese grid mix (Figure 3). The products obtained from its valorisation have an economic and environmental value and hence have an impact on market dynamics. The production of compost and mulch impacts the demand for conventionally used chemical fertilisers, while the production of electricity from biomass has an impact on the national energy mix

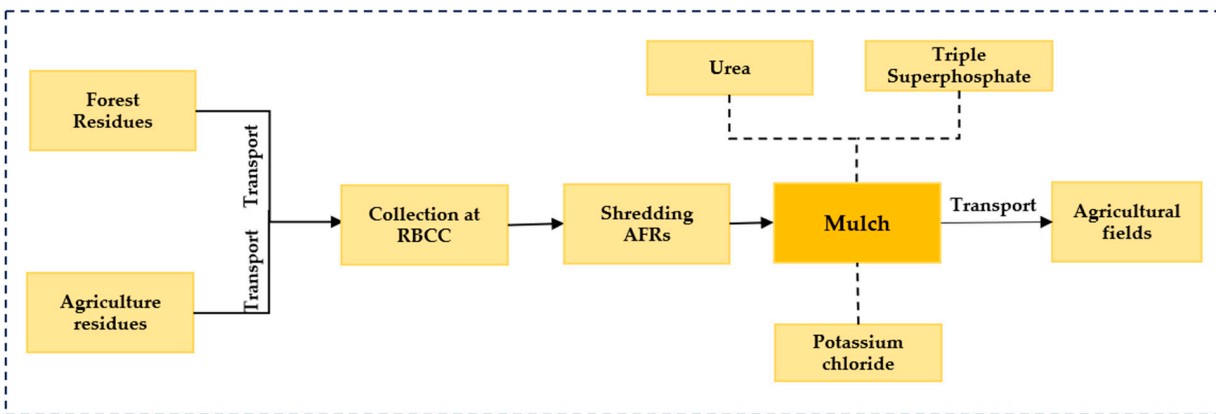

**Figure 1.** Definition of the system boundaries in the ARF mulching process.

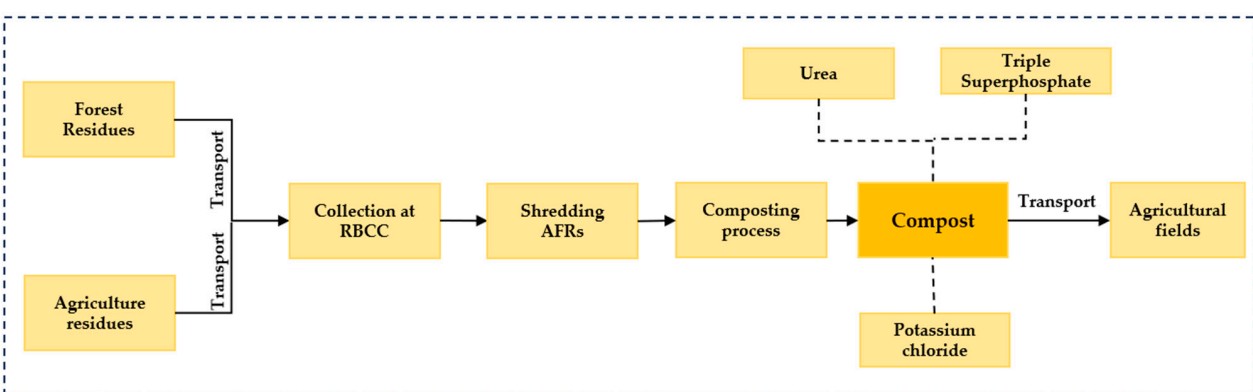

**Figure 2.** Definition of the system boundaries in the ARF composting process.

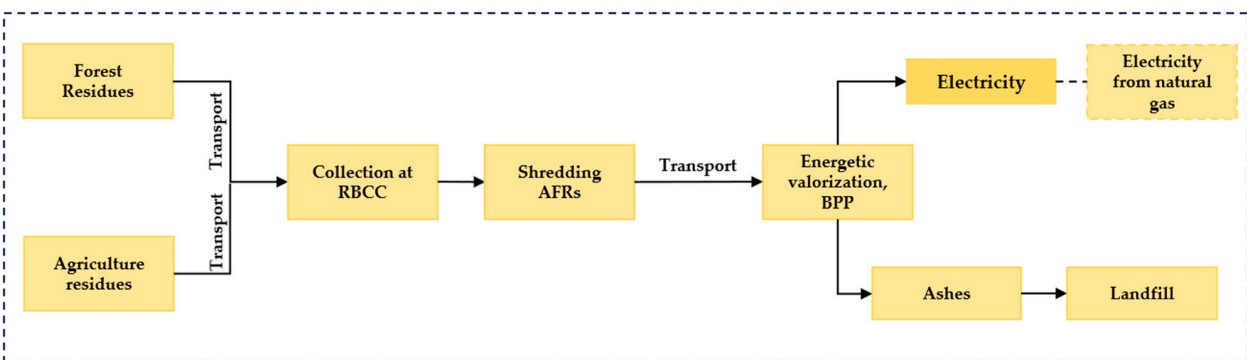

**Figure 3.** Definition of the system boundaries in the valorisation of ARF for electricity production.

### 2.1.2. Functional Unit

All the input and output loads are referred to as the functional unit (FU) of 1 tonne (37% moisture content, according to experimental procedures) of AFRs collected at the RBCC of Bodiosa.

### 2.2. *Inventory Data*

The most complex phase within a LCA investigation concerns the LCI, wherein the imperative task is the acquisition of data that closely align with real-world processes. This rigorous data-collection process is essential to ensure that inventory analysis and impact assessment within the study remain consistent with the study's objectives. The collected data must exhibit temporal and geographic relevance and technological representativeness.

Within the scope of this study, a portion of foreground data was derived from laboratory analyses pertaining to biomass and compost. Subsequently, extra data were sourced from the Biomass Power Plant and Ecoinvent, 2021 database.

### 2.2.1. Production of Forest Residues

In some of Portugal's northern districts, pine accounts for more than half of the total forest area, while eucalypts are favoured in the coastal districts of the northern regions [25]. Given that the most present species detected in the RBCC is maritime pine (*Pinus pinaster*, Aiton), the impacts associated with the production of forest residues were assessed [26], where an economic criterion has been used to allocate the burdens to Roundwood, industrial wood, and residual wood produced from maritime Portuguese pine. To relate the outputs to the FU of the study, a density of 0.55 tonnes/m$^3$ was used. Due to the lack of data, the production of agricultural residues was neglected, assuming they have no economic value and no burdens allocated.

### 2.2.2. Transfer of AFRs to the RBCC

The following process is the collection and subsequent delivery of residual biomass to the RBCC. Particularly, it is essential to know the operating time of the vehicles that are conventionally used to transport the biomass: fuel consumption and associated emissions are tightly connected to this parameter. These collection centres for any type of biomass should be designed over an influence area of 10 km [27]. This is the maximum distance that a generic user is set to travel, beyond which the perceived utility of the service decreases. To measure the necessary time parameter, analysis through a procedure implemented in QGIS has been executed. In particular, the OpenRouteService (ORS) Tools plugin (2022 openrouteservice) was used, useful in accessibility studies of a service. Through it, a portion of the territory from which users can reach the RBCC, travelling less than 10 km on the OpenStreetMap road network with an agricultural vehicle, has been delimited. The advantage consists of a more realistic distance assessment than what would have been obtained considering a simple circular buffer.

In the identified area, the forest and agricultural soils were determined by selecting the corresponding references in Corine Land Cover (CLC) 2018, the land cover map developed by Copernicus (Supplementary Materials—Figures S1 and S2). Depending on the type and (approximate) quantity of plant species found during the samplings, the CLC entries most representative of the territory under examination were selected. In this way, different portions of the territory were determined with their associated area extension.

A residual biomass productivity rate (tonne/ha) was assessed on an annual scale for different plant species (Supplementary Materials—Table S1), according to Fernandes and Costa [28]. The values were averaged between the different species belonging to each CLC field. Multiplying the productivity rate by the areal extent of each land portion results in the amount of residual biomass produced. The total amount is 3877.88 tonnes, with 76% consisting of forest residues and 24% consisting of agricultural waste. Data were then expressed in proportion to the FU of the study (i.e., 1 tonne of residual biomass) to spread the residual biomass production over the territory. To identify an area of the CLC region, the centroid that represents it in its entirety is used. Displacements are assumed to start. Under the assumption that all the biomass produced in each homogeneous zone is loaded onto a farm vehicle (with a load capacity of 1 tonne) and transported to the RBCC, the time required to complete each journey was estimated through ORS Tools. Through a calculation considering the sum of each fraction of biomass transported and the respective time required divided by the capacity of the transport vehicle, it was possible to estimate an average time for the delivery of residual biomass. A weighted average is used to calculate the transport time expressed in minutes. The product of the time required for the transport of each load multiplied by the biomass fraction transported is added up; then, everything is divided by the transport capacity of the vehicle, expressed in tonnes. In conclusion, it takes 7.48 min to collect the above-mentioned amount of residual forest biomass at RBCC,

while it takes 2.42 min for agricultural biomass. Forest biomass is produced in significantly wider areas spread throughout the territory. Furthermore, since its productivity rate is, in most cases, considerably lower than for residual biomass of agricultural origin, the delivery of 0.76 tonnes of forest residues requires them to be picked up from various points around the territory, with a greater impact in terms of transport. All data is shown in Table 1.

**Table 1.** Inventory data for production and transportation of AFRs.

| | Process/Material | Quantity | Unit of Measure |
|---|---|---|---|
| **Inputs** | Maritime pine wood at forest road [26] | 1.39 | $m^3$ |
| | Transportation of forest residual biomass | 7.48 | Min |
| | Transportation of agricultural residual biomass | 2.42 | Min |
| **Outputs** | Forest residues at RBCC | 0.76 | Tonnes |
| | Agricultural residues at RBCC | 0.24 | Tonnes |

### 2.2.3. Shredding of AFRs

The subsequent process is biomass shredding to make it suitable for the following treatments. Biomass is taken through an excavator with a grapple and fed into a crushing machine with a hammer mill, whose productivity is 20 tonne/h. Three minutes are required to shred 1 tonne, which is the operating time of both machines (Table 2). There is a 2% loss of dry matter in the biomass as a result of shredding [29,30].

**Table 2.** Inventory data for the biomass shredding.

| | Process/Material | Quantity | Unit Measure |
|---|---|---|---|
| **Inputs** | Residues at RBCC | 1 | Ton |
| | Biomass shredding | 3 | Min |
| **Outputs** | Shredded biomass at RBCC | 0.980 | Tonne |

After biomass shredding, the steps characterising the three processes under consideration are different; hence, the analysis is performed separately.

### 2.2.4. Composting

The AFR's composting process was carried out without pile-turning or irrigation. The pile had a cone shape with 2 m base diameter and 1.2–1.5 m height. The final volume and weight of the pile were evaluated based on direct measurement of the pile dimensions and the compost density analysis, respectively.

A compost production index, $I_c$, was assessed to determine the amount of compost production from the FU, leading to 0.614 tonnes of compost. Since emissions were not monitored during the composting processes, results were from the composting process performed on a similar biomass [31], where the data were from the Ecoinvent database [32]. From the compost nitrogen, potassium, and phosphorous content (the first two evaluated in laboratory studies, the latter from the literature), the amount of avoided fertilisers was estimated. The fertilisers were urea, potassium chloride (KCl), and triple superphosphate (TSP). This choice is supported by the trends in demand/consumption of traditional fertilisers in Portugal [33]. The equivalence in terms of nutrients between compost and mineral fertilisers was established based on the respective nutrients' bioavailability. The amount of N contained in urea is 450 g/kg, and 56% is available for plants. Regarding TSP, its P content is 210 g/kg, of which 86.5% is available for plants [34]. The amount of K in the KCl is 498 g/kg, and 67.5% of this K is available for plants [35]. The total phosphorus of compost (in dry mass) is 0.24% [36]. The nutrients in the compost are supposed to be fully

bioavailable to plant species, resulting in a lack of emissions due to unabsorbed ones. If compost is stable, an essential requirement for its use in agriculture, it generates minimal emissions after its application, which were neglected in this study. Also neglected, as they are difficult to quantify, are the benefits generated by compost after its use. In Table 3, input and output data about the composting processes are shown.

**Table 3.** Inventory data for composting process.

|  | Process/Material | Quantity | Unit of Measure |
|---|---|---|---|
| **Inputs** | Shredded biomass | 0.980 | tonnes |
|  | Oxygen | 267.000 | kg |
|  | Land use | 18.896 | $m^2/y$ |
|  | Water | - | $m^3$ |
|  | Transportation to fields | 6.365 | min |
| **Outputs** | Compost | 0.614 | tonnes |
|  | $NH_3$ | 0.020 | kg |
|  | $CO_2$ | 404 | kg |
|  | $N_2O$ | 0.051 | kg |
|  | $CH_4$ | 0.047 | kg |
|  | NMVOC | 0.019 | kg |
|  | $H_2O$ | 0.189 | $m^3$ |
| **Avoided products** | Urea | 9.542 | kg |
|  | TSP | 3.176 | kg |
|  | KCl | 5.117 | kg |

2.2.5. Mulching

Shredded residual biomass can be used directly in agricultural fields as fertiliser. Its nutrient content is highly variable depending on its composition. After being shredded, the AFRs are transported to agricultural fields; depending on their nutrient content, they are used to replace conventional fertilisers. To evaluate the transport time, the same procedure used for the composting process has been applied.

The nitrogen content of the biomass, as well as the moisture content, was derived from the analyses carried out on the sample taken at RBCC. Since the characterisation of AFRs at RBCC (at various times of the year) showed that the species most frequently encountered is maritime pine, the phosphorus and potassium content were approximated to that of the mentioned species [33]. Data are reported in Table 4.

**Table 4.** Composition of the mulch.

| Nutrients | Amount (g/kg$_{db}$) |
|---|---|
| **N** | 6.31 |
| **P** | 0.23 |
| **K** | 1.38 |

The assessment of the bioavailability of such nutrients and their partial release in form of emissions to soil, water, and atmosphere is mentioned in previous research [33]. Because of the lack of specific data concerning forest biomass, emission factors are based on crop residues, and they express the percentage of nitrogen that is lost through volatilisation or leaching:

A total of 1.25% directly emitted as $N_2O$;
A total of 10.5% emitted as $NH_3$;
A total of 0.7% emitted as $NO_x$;
A total of 10% emitted as $NO_3^-$.

Moreover, indirect $N_2O$ emissions from $NH_3$ and $NO_x$ volatilisation and from $NO_3^-$ leaching were considered:
A total of 0.010 kg $N_2O$ per kg of volatilised $NH_3$ and $NO_x$;
A total of 0.025 kg $N_2O$ per kg of $NO_3^-$ leached.

The P available for plants can range from 5 to 22% of the total P contained in the forest biomass residues. Considering the mean value of this range (13.5%), the remaining part (86.5%) is emitted to the soil. The K available for plants considered in this study is 17.4% of the total K in the forest biomass residues. The remaining K is emitted to the soil. According to the amount of N, P, and K available for plants, the amount of avoided fertilisers is assessed, following the same procedure used for composting. Results are presented in Table 5.

**Table 5.** Inventory data for mulching.

|  | Process/Material | Quantity | Unit of Measure |
|---|---|---|---|
| **Inputs** | Shredded biomass | 0.980 | tonnes |
|  | Transportation to fields | 10.16 | min |
| **Outputs** | Mulch | 0.980 | tonnes |
|  | $N_2O$ | 0.073 | kg |
|  | $NH_3$ | 0.473 | kg |
|  | $NO_x$ | 0.032 | kg |
|  | $NO_3^-$ | 0.451 | kg |
|  | P- emitted to soil | 0.142 | kg |
|  | K- emitted to soil | 0.815 | kg |
| **Avoided products** | Urea | 13.809 | kg |
|  | TSP | 0.122 | kg |
|  | KCl | 0.510 | kg |

2.2.6. Energetic Valorisation

Currently, the biomass deposited at the RBCC is transported to the *Biomass Power Plant* of Mundão. Here, from the valorisation of residual forest biomass collected from a maximum distance of 150 km, electricity is produced and sold to the national electricity grid. The inventory analysis for this process was carried out based on the operating data of the power plant collected after technical visits. The AFRs are transported to the plant by lorries in the Euro5 category, assuming an average age of 10 years. Then, biomass is conveyed to the boiler, which operates at a temperature of around 700 °C, through a belt. The composition of the biomass fed into the boiler is as follows:

A total of 40% biomass similar to that coming from RBCC;
A total of 20% woody trunks that cannot be used in other sectors (e.g., industrial);
A total of 20% leaves and branches;
A total of 20% woody biomass with a high calorific value.

The heat generated by combustion is used to raise the temperature of a fluid that expands in a turbine with an installed capacity of 15 MW. The electricity production amounts to 15 MWh, while 1 MWh of it is required to electrically support all the plant's components. This resulted in an electrical energy net production of 14 MWh.

The water used in the plant comes from current wells. To meet the quality requirements of the feed water to the boiler, a treatment process (demineralisation) is necessary. The demineralisation plant consists of microfiltration and two osmosis units to reduce the amount of dissolved salts and an electro-deionisation unit to reduce the ion concentration. Actual water usage linked to the combustion process is 2.5 $m^3$/h.

To comply with environmental regulations, exhaust gases leaving the steam generator first pass through a cyclone and then through a bag filter. In the cyclone, a high-speed rotating air stream removes the denser particles, which are collected at the bottom. The subsequent bag filter removes the particles suspended in the flue gas, preventing their release into the atmosphere. To assess conformity with legal limits, emissions are periodically monitored. The values recorded during the 20 November 2020 measurements are shown in Table 6.

**Table 6.** Emissions from Biomass Power Plant on 20 November 2020.

| Parameter | Concentration (mg/$m^3$) $_{PTN}$ | Volumetric Flow ($m^3$/h) $_{PTN}$ | Emissions (kg/h) |
|:---:|:---:|:---:|:---:|
| CO | 48 | | 3.39 |
| $NO_x$, expressed as $NO_2$ | 187 | | 13.2 |
| Total particles | 14 | | 1.00 |
| Organic carbon compounds, expressed as Total C | 22 | 70,526 | 1.55 |
| $CO_2$ | 235,621 | | 16,617 |

The amount of particles produced for the FU was assimilated to the value of the ashes. Due to the lack of data on their final destination, their disposal in landfill is assumed, and due to the small quantity of ashes produced from the FU, impacts associated with their transportation to the landfill are not considered. Nevertheless, the government indicates that natural gas electricity generation will be maintained until at least 2040 [37]. The production of electricity from this plant will consider avoiding emissions from Portuguese grid mix electricity. A mass-allocation criterion was used to express all inputs and outputs of the process in relation to the FU of the study. All loads were multiplied by a factor of 40% to consider only those associated with the AFRs from RBCC (Table 7).

**Table 7.** Inventory data for energetic valorisation.

| | Process/Material | Quantity | Unit of Measure |
|:---:|:---:|:---:|:---:|
| **Inputs** | Shredded biomass | 0.980 | tonnes |
| | Transportation to Biomass Power Plant | 17.3 | tonnes/km |
| | Water (treatment) | 98.0 | kg |
| **Outputs** | Electricity | 0.549 | $MWh_{el}$ |
| | CO | 0.133 | kg |
| | $NO_x$, as $NO_2$ | 0.517 | kg |
| | TP | 0.039 | kg |
| | COV | 0.061 | kg |
| | $CO_2$ | 651 | kg |
| | Ashes (landfilling) | 24.8 | kg |
| **Avoided products** | Electricity, from country mix | 0.549 | $MWh_{el}$ |

### 2.3. Impact Assessment

The impact assessment phase of LCA is aimed at evaluating the significance of potential environmental impacts using the LCI results. In general, this process involves associating inventory data with specific environmental impact categories and category indicators (Supplementary Materials—Table S2), thereby attempting to understand these impacts [22]. Impacts assessed with LCA are potential, not real, since they describe the life cycle impact of a reference flow used to describe a FU. Impacts associated with secondary flows produced by the same process are not considered. Furthermore, they are based on inventory data integrated in space and time; indeed, they often occur in different locations and time horizons.

The chosen method, CML-Baseline, was developed in 2001. It is a European method, and its approach is midpoint. The results can be normalised, but neither weighting nor summation is provided. The impact categories considered are abiotic depletion (AD), abiotic depletion associated with fossil fuels (ADff), global warming (GW), ozone layer depletion (OLD), human toxicity (HT), freshwater aquatic eco-toxicity (FE), marine ecotoxicity (ME), terrestrial ecotoxicity (TE), photochemical oxidation (PO), acidification (A), and eutrophication (E).

### 3. Results

The results obtained in the LCA analysis are shown in Figures 4–6. The data tables with the quantified emissions are described in Supplementary data. In each graph, the negative environmental impact from the process is shown in the positive sector of the diagram, and avoided impacts are shown in the negative sector. The concept of avoided impacts arises from the extended system boundaries that include the prevented products and their emissions.

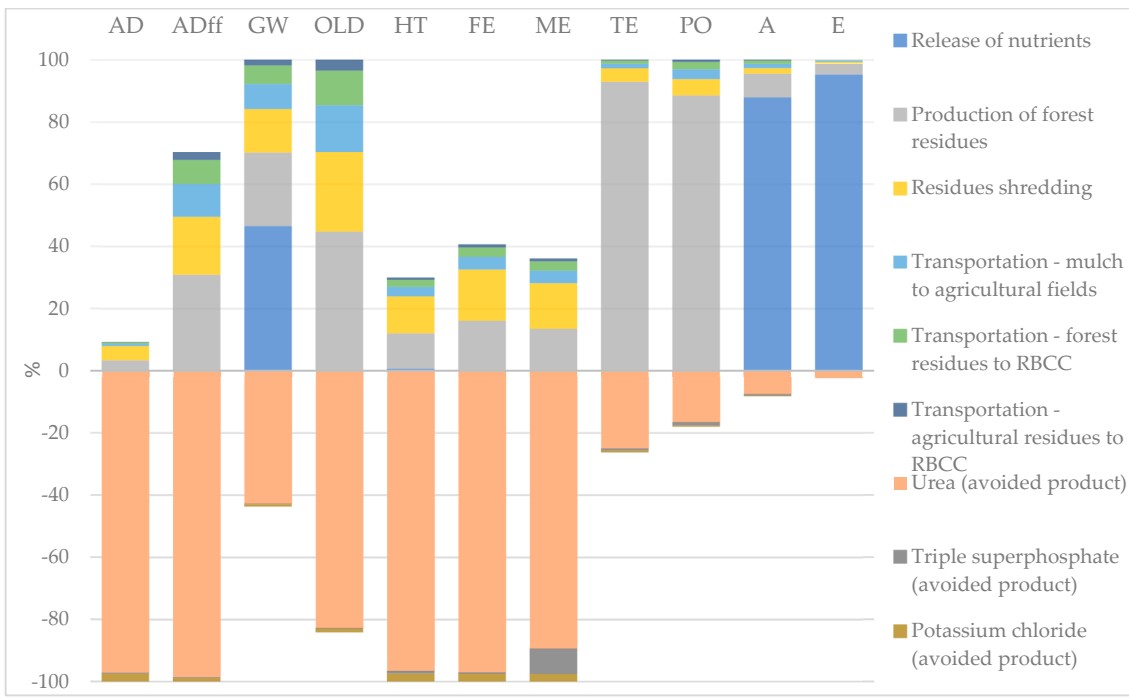

**Figure 4.** Impact assessment results for mulching (abiotic depletion (AD), abiotic depletion associated with fossil fuels (ADff), global warming (GW), ozone layer depletion (OLD), human toxicity (HT), freshwater aquatic eco-toxicity (FE), marine ecotoxicity (ME), terrestrial ecotoxicity (TE), photochemical oxidation (PO), acidification (A), and eutrophication (E)).

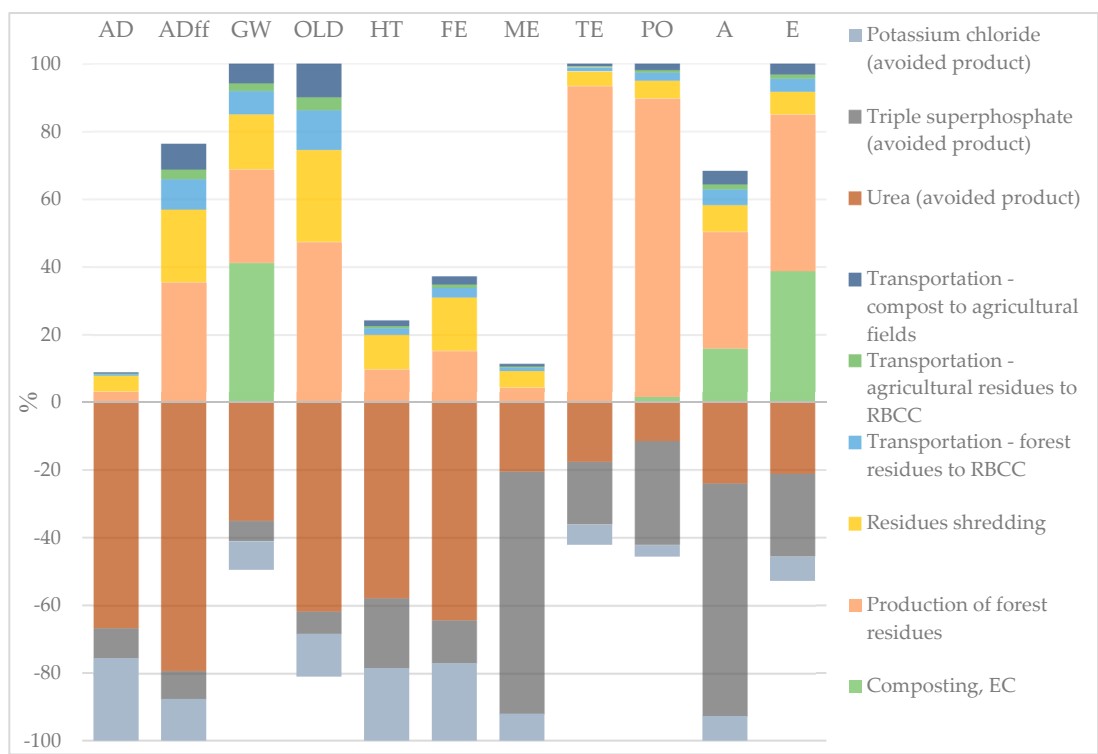

**Figure 5.** Impact assessment results for composting (abiotic depletion (AD), abiotic depletion associated with fossil fuels (ADff), global warming (GW), ozone layer depletion (OLD), human toxicity (HT), freshwater aquatic eco-toxicity (FE), marine ecotoxicity (ME), terrestrial ecotoxicity (TE), photochemical oxidation (PO), acidification (A), and eutrophication (E)).

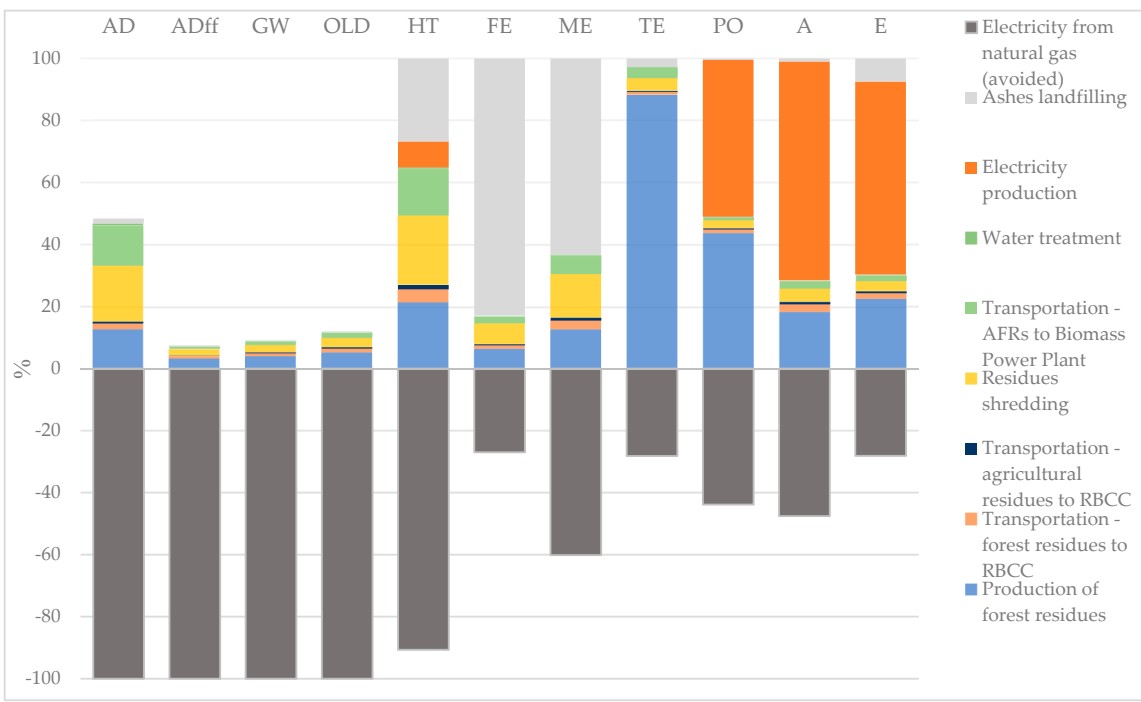

**Figure 6.** Impact assessment results for energetic valorisation, Biomass Power Plant (abiotic depletion (AD), abiotic depletion associated with fossil fuels (ADff), global warming (GW), ozone layer depletion (OLD), human toxicity (HT), freshwater aquatic ecotoxicity (FE), marine ecotoxicity (ME), terrestrial ecotoxicity (TE), photochemical oxidation (PO), acidification (A), and eutrophication (E)).

Figure 4 shows the impacts for each stage of mulch production from AFRs (the results of the characterisation of the LCIA results for mulching are presented in Supplementary Materials—Table S3). Forest biomass production/preparation involves activities related to the use of equipment and machinery for forest maintenance that ensure the removal of undesirable trees or tree branches, enhancing the tree's structure and guiding fresh, healthy growth. As shown in the graphic, these activities have a significant impact, mainly in terms of terrestrial ecotoxicity and photochemical oxidation, due to the emissions of cypermethrin and trichlorfon to the soil and formaldehyde and mercury to air for the first impact category. The most contributing substances to photochemical oxidation are carbon monoxide, toluene, and formaldehyde. The same behaviour is seen in the other stages, whose activities are related to the use of fossil fuels, mulch transportation to agricultural fields, and residue shredding, even considering the relatively short distances considered in the study. Biomass preparation also considerably contributes to abiotic depletion, global warming, and ozone layer depletion. The use of fossil fuels impacts highly on the abiotic depletion associated with fossil fuels, while the emissions of carbon dioxide, nitrous oxide, and methane impact global warming. Ozone layer depletion impacts arise from the emissions of Halon 1301 and Halon 1211. The release of nutrients, as a consequence of mulch application in the fields, affects global warming due to nitrous oxide emissions. Ammonia and nitrogen oxides are the most contributing substances to acidification, and their combination with phosphorus and nitrate release contributes to the process's impact on eutrophication being significantly high. In terms of avoided products, the main benefits result from avoiding urea fertiliser production, which is the main fertiliser, as it has a high nitrogen concentration and low cost. The benefits are reflected in all impact categories considered, but mainly abiotic depletion, abiotic depletion due to fossil fuels, human toxicity, freshwater aquatic eco-toxicity, and marine ecotoxicity, with a contribution higher than 89%. For the first two categories, the reduced impacts derive from the avoided extraction of raw resources such as natural gas, while for the last three, they derive mainly from avoided emissions of chromium, nickel, copper, and beryllium.

The impact assessment of the AFR composting process is shown in Figure 5 (the results of the characterisation of the LCIA results for composting are presented in Supplementary Materials—Table S4). As in the previous treatment option, forest residue production, resulting from pruning and forest fuel management, gives a considerable contribution to all environmental impact categories, especially to terrestrial ecotoxicity and photochemical oxidation (see mulching), responsible for 88–93% of the overall impacts. Simultaneously, the shredding of AFRs affects all categories, with the largest contribution to ozone layer depletion (27%) and abiotic depletion due to fossil fuels (21%). Oil extraction for subsequent use in the shredding machine contributes the most to abiotic depletion due to fossil fuels, while natural gas and coal contributions are slightly lower. The most important factor influencing ozone layer depletion is Halon 1301 emissions to air. The same two categories are also affected by transportation process-related impacts (oil and methane are the most impactful substances, even in this situation).

The composting process, in which organic matter decomposes and releases emissions, as quantified in the inventory analysis, is shown to be negligible for eight out of the eleven impact categories under study. The emission of nitrous oxide strongly impacts global warming (41%), making the composting process the most important in this category. At the same time, the process has a 16% influence on category acidification due to ammonia emissions to air and a 39% influence on category eutrophication due to ammonia, phosphate, and nitrous oxide release.

The avoided impacts, related to the whole urea production, contribute more than triple superphosphate and potassium chloride production. The main benefits of its avoided production reflect abiotic depletion and abiotic depletion due to fossil fuel categories and show slight benefits to ozone layer depletion, human toxicity, and freshwater aquatic ecotoxicity. The benefits in the abiotic depletion category are due to the non-extraction of substances such as tellurium and molybdenum, which can be used in the production

of fertilisers, as well as abiotic depletion due to fossil fuels because of the avoidance of natural gas, oil, and coal consumption, which are used in the production of the energy required for the fertiliser synthesis. The decrease in triple superphosphate production is the main contributor to the avoided impacts on terrestrial ecotoxicity, photochemical oxidation, acidification, and eutrophication. They are related to the avoided emissions of cypermethrin for terrestrial ecotoxicity, sulphur dioxide for photochemical oxidation and acidification, and phosphorus—leached in water—for ecotoxicity.

The graphic in Figure 6 characterises the impact assessment for the energetic valorisation process (characterisation data of the LCIA results for energetic valorisation are presented in Supplementary Materials—Table S5). AFR transportation has a negligible environmental impact. The transport of residual biomass from the RBCC to the Biomass Power Plant contributes to 13–15% of abiotic depletion and human toxicity due to the short spatial distance. Again, the forest residue preparation has the largest impact on terrestrial ecotoxicity (88%) and contributes to 44% of photochemical oxidation. For the other impact categories, its influence is minimal. The release of substances such as vanadium, nickel, molybdenum, and arsenic into water from the ash disposal in landfills contributes to human toxicity (27%). The same substances, combined with copper and zinc, also impact on freshwater aquatic ecotoxicity (63%) and marine ecotoxicity (83%). Electricity production, causing emissions during the residual biomass combustion process, has a predominant impact on photochemical oxidation, acidification, and eutrophication categories (51–71%), with the former being caused by nitrogen dioxide emissions and mainly carbon monoxide release into the air. Without electricity production from natural gas, which is a fossil fuel, advantages arise for all impact categories. Those most positively affected are the abiotic depletion category as a result of the avoided extraction of tellurium and copper and the abiotic depletion associated with the fossil fuels category because of the avoided use of natural gas. Likewise, a reduction in global warming benefits from the lower emissions of carbon dioxide and methane, while advantages for the ozone layer depletion category are achieved by the avoided emissions of Halon 1211 and HCFC-22.

The total impacts, those positive and avoided, are summarised in Table 8. The major benefits of applying valorisation techniques to AFRs are the decrease in the abiotic depletion associated with fossil fuels, as well as the abiotic depletion of other natural resources. Energetic valorisation significantly decreases the impacts of global warming and ozone layer depletion. On the other hand, human toxicity, freshwater aquatic ecotoxicity, and marine ecotoxicity benefit from the use of AFRs valorisation through composting or mulching. It can also be seen that the energetic valorisation of 1 tonne of ARFs decreases the release of carbon emissions by about 218 kg $CO_{2\,eq}$, while composting and mulching of 1 tonne of ARFs creates 1.88 and 2.33 kg $CO_{2\,eq}$ emissions, respectively.

Figure 7 shows the impact category comparison for the three AFRs valorisation options under study. It is possible to understand which of the processes carries the highest impact in the different considered categories. The benefits of generating electricity from biomass instead of the Portuguese grid mix are reflected in Electricity Production's better performance in abiotic depletion due to fossil fuels, global warming, and ozone layer depletion categories, as mentioned above. Thermal valorisation processes are shown to be worse in comparison to biological valorisation processes (composting and mulching) for the categories of human toxicity, freshwater aquatic eco-toxicity, marine ecotoxicity, terrestrial ecotoxicity, and photochemical oxidation. Conversely, mulching emerges as the process with the worst environmental performance for the impact categories of acidification, eutrophication, and global warming. In particular, it can be observed that the impact in terms of acidification and eutrophication is much higher than in the case of composting due to the higher percentage of nitrogen and phosphorous that is not absorbed by the plants and is released into the soil and water.

**Table 8.** Impact assessment—comparison of processes.

| Impact Category | Unit | Composting | Mulching | Electricity Production |
|:---:|:---:|:---:|:---:|:---:|
| **AD** | kg $Sb_{eq}$ | $-2.54 \times 10^{-4}$ | $-2.51 \times 10^{-4}$ | $-3.67 \times 10^{-5}$ |
| **ADff** | MJ | $-8.87 \times 10$ | $-1.28 \times 10^2$ | $-3.74 \times 10^3$ |
| **GW** | kg $CO_{2\ eq}$ | $1.80 \times 10$ | $2.33 \times 10$ | $-2.18 \times 10^2$ |
| **OLD** | kgCFC-11$_{eq}$ | $6.70 \times 10^{-7}$ | $5.92 \times 10^{-7}$ | $-2.76 \times 10^{-5}$ |
| **HT** | kg1,4-DB$_{eq}$ | $-1.22 \times 10$ | $-9.72$ | $6.89 \times 10^{-1}$ |
| **FE** | kg1,4-DB$_{eq}$ | $-5.52$ | $-4.97$ | $1.50 \times 10$ |
| **ME** | kg1,4-DB$_{eq}$ | $-3.77 \times 10^4$ | $-8.99 \times 10^3$ | $5.86 \times 10^3$ |
| **TE** | kg1,4-DB$_{eq}$ | $5.43 \times 10^{-2}$ | $6.95 \times 10^{-2}$ | $7.15 \times 10^{-2}$ |
| **PO** | kg $C_2H_{4\ eq}$ | $9.64 \times 10^{-3}$ | $1.45 \times 10^{-2}$ | $2.01 \times 10^{-2}$ |
| **A** | kg $SO_{2\ eq}$ | $-6.22 \times 10^{-2}$ | $8.07 \times 10^{-1}$ | $1.92 \times 10^{-1}$ |
| **E** | kg $PO_4^{-}{}_{eq}$ | $2.50 \times 10^{-2}$ | $6.86 \times 10^{-1}$ | $7.76 \times 10^{-2}$ |

Abiotic depletion (AD), abiotic depletion associated with fossil fuels (ADff), global warming (GW), ozone layer depletion (OLD), human toxicity (HT), freshwater aquatic eco-toxicity (FE), marine ecotoxicity (ME), terrestrial ecotoxicity (TE), photochemical oxidation (PO), acidification (A), and eutrophication (E).

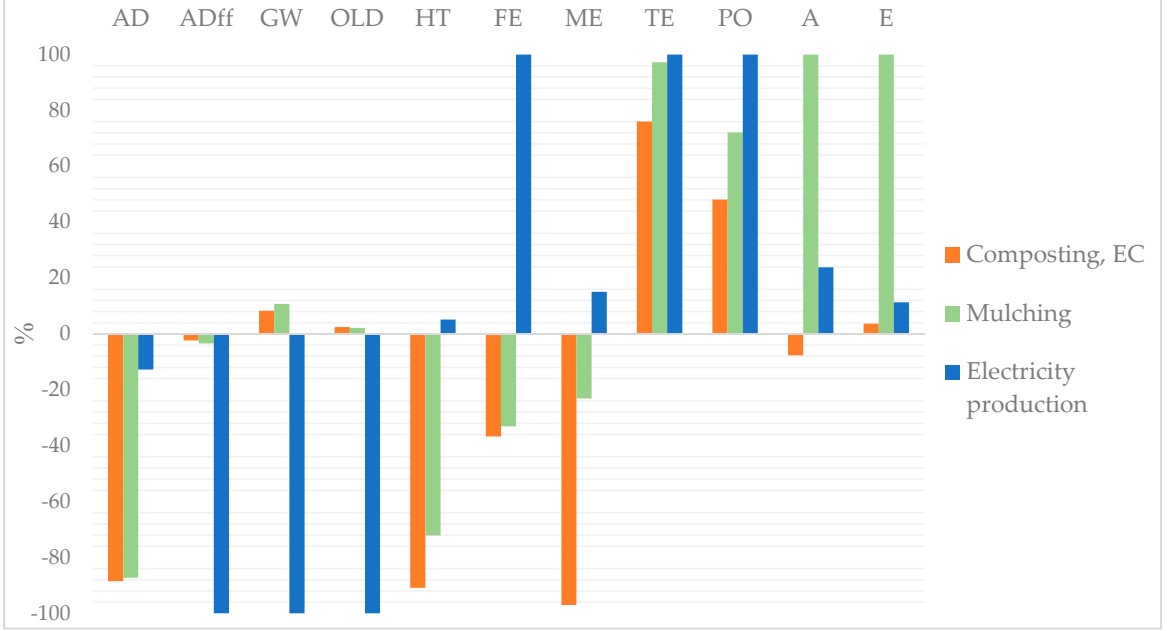

**Figure 7.** Impact assessment, comparison between processes (abiotic depletion (AD), abiotic depletion associated with fossil fuels (ADff), global warming (GW), ozone layer depletion (OLD), human toxicity (HT), freshwater aquatic eco-toxicity (FE), marine ecotoxicity (ME), terrestrial ecotoxicity (TE), photochemical oxidation (PO), acidification (A), and eutrophication (E)).

In nine impact categories, with the exception of ADff and OLD, the composting processes appear to be the least impactful. The quality of the final compost, in terms of nutrients, is the key factor that generates the slight differences between the processes.

## 4. Discussion

According with the LCA results the, considering the assumptions and limitations of the study, it is possible to identify which treatment is the most advantageous. The lack of previous literature comparing the environmental impacts of these three processes using

an LCA underlines the importance of this research. This study showed that composting has the lowest overall impact, except for abiotic depletion due to fossil fuels and ozone layer depletion, even compared to mulching. Mulching has a very different environmental performance due to both the low nutrient content of the biomass produced and the release of a large amount of nutrients as emissions. Nevertheless, composting and mulching are shown to be the processes with a higher global warming impact. Other research [38] has also assessed that the decomposition emissions during composting have a large environmental impact on global warming potential, acidification, and eutrophication. This is in line with other studies reported [39], even though a different impact assessment method was used. It was found that ammonia emission contributes to 56% of the eutrophication potential and 32% of the acidification potential. Serafini et al. (2023) performed an important literature review about the environmental impacts of the composting process and presented similar results [40]. These authors report that out of all the effect categories, emissions from the use of fossil fuels and electricity for the process outcome were primarily linked to 50% of the impacts, namely, global warming potential, ozone layer depletion, human toxicity, abiotic depletion, and fossil depletion potential. High-energy content material-consuming processes typically rate higher in these categories.

The advantage of composting over the other treatments is mainly due to its use as an agricultural soil improver. In fact, this type of valorisation allows AFRs to be transformed into a new product with a commercial value because it allows those who produce it to save on synthetic fertilisers, and with an environmental value since it allows important nutrients capable of enriching the geosphere and biosphere to be re-transferred to the soil. As in mulching, the organic carbon may be retained in the soil, mainly in the carbon content in fine roots of subsoil, which is very important for the microbial biomass in the soil [41,42].

Also, the production of compost is possible without the control of physical and chemical parameters throughout the process, nor the addition of water or the turning of the pile, with minimum intervention and consequent increase in energy demand. This allows small landowners to independently manage pruning, agricultural crop residues, and other biomass residues, producing a final product of good quality with the advantage of local utilisation that is used in the same agricultural fields or forestry lands where it is produced. Along with the material recovery and carbon sequestration, further benefits are obtained from the reduction in fuel consumption, with its associated emissions, caused by the transport of biomass to the RBCC. Due to the absence of rotation, anaerobic conditions within the heaps must be avoided. In fact, the anaerobic degradation of organic matter produces significant amounts of methane that would be emitted into the atmosphere with consequent significant impacts. To avoid this risk, it is preferable to build smaller piles that promote the entry of air and allow composting in aerobic conditions. This good practice also reduces the possibility of starting fires, favoured by the heat generated during the process and the presence of methane as a reaction fuel. However, the significant importance of existing RBCCs for these residues cannot be denied. Indeed, they represent the best solution for the proper disposal of AFRs when they are originated from the management of public green areas; forest management; or when the composting technique, as mentioned above, cannot be utilised.

AFR production and preparation—that is, harvesting and biomass management on-site—represented the major sources of negative impacts, namely, terrestrial ecotoxicity and photochemical oxidation, as well as global warming. The same was reported by other researchers [43]. The increase in productivity and environmental efficiency should be emphasised with the use of mechanised methods and good-quality equipment and tools. Some fully mechanised felling and processing with harvester methods (tracked/crawler harvester) should be used. When it is not possible, motor–manual felling and processing must be applied. The last option is more versatile once it overcomes the limitations of the former, namely, the narrow focus of application and the inability to be used in rainy periods. In fact, motor–manual equipment, using electro-mechanical chainsaws and skidders, it is easy to operate, as it features broader labour qualification, requires only small

investments, can sustain periods of low activity, and can be operated under a large range of site conditions [44].

ARF transportation has significant environmental impacts on its overall management due to the emissions of carbon monoxide, nitrogen oxides, sulphur dioxide, and volatile non-methane organic compounds and particulate (PM 2.5) [45]. As referred in the literature, many eco-friendly options are practical and implementable in the near future, some of them easier and faster to implement [46]. A suggestion to this may be fuel and engine innovations in infrastructure maintenance and route planning or other innovations and methods to lower emissions, like larger, more powerful vehicles and conduction strategies for fuel economy and a decrease in polluting emissions.

Studying the treatment of AFRs from a circular economy perspective allows an increase in their value. The management of this biomass has always been seen as a problem increasing the managing costs of forest or agriculture activities, but it is possible to change this paradigm. Today, in fact, it is possible to approach AFRs as an opportunity with positive economic and social consequences because of their intrinsic value. But also, environmental benefits. LCA is an irreplaceable tool due to its standards, reliability, and accuracy, which allow two results to be achieved. On the one hand, it is possible to objectively quantify all the impacts associated with one technique to compare it with the impacts related to another. In addition, it allows us to identify the impacts related to each single process and recognise where action can be taken to reduce the negative ones. It provides all the elements for an objective impact assessment. These results are a useful tool for local public decision makers and should be used to guide and define the best management policies for AFRs.

Five main guiding concepts should be used to develop national strategies, incentives, rules, and policies that promote profitable, environmentally safe, and economically sustainable agroforestry, namely: (1) effective resource utilisation for sustainable agriculture; (2) conservation, protection, and enhancement of natural resources promoting sustainability; (3) employment of sustainable agriculture to improve social well-being and rural livelihoods; (4) adoption of sustainable agriculture to enhance the resilience of communities and ecosystems towards market volatility and climate change; and finally, (5) that the sustainability of natural and human systems is essentially under the control of good governance [47].

It is generally recognised that the current energy model is unsustainable due to the enormous dependence on fossil fuels, in particular oil. It is therefore necessary to review and reorient energy policies, with emphasis on the diversification of energy sources and the growing use of renewable energies. The forestry and energy policies are intended to enhance the exploitation and rational use of forest resources, particularly the biomass component, optimising the various processes involved in the cleaning, collection, sorting and storage of this resource, with the creation of RBCCs. Thus, it will be possible to foster the emergence of a regional market, mobilising forest producer organisations, operators, service providers, and municipalities, as well as promoting the use of equipment supported by technological solutions adapted to the regional reality of a specific area.

The use of residual biomass for energy production, as mentioned above, has positive aspects, namely reducing greenhouse gas emissions. However, it implies the existence of power plants in various locations, so it can be sustainable from the point of view of transport. Simpler strategies that require fewer resources, with easy local implementation, can mobilise a more sustainable valorisation of the residual fraction of biomass. In fact, in addition to adding nutrients, mulching is a particularly effective technique for preserving soil moisture, temperature and structural stability and boosting crop output, creating favourable microclimatic conditions adjacent to the plant. This technique is less expensive than the other AFRs valorisation under study since fewer pre- and post-plantation maintenance tasks are required. Otherwise, composting requires more activities and is time-consuming for preparation and application beneath the top layer of the soil, near the plant roots, which is much richer in nutrients and organic matter; even so, it can be used locally. If the proper knowledge and techniques are spread among the local players, supported by

financial, national, or regional programs, residual biomass can be reintroduced into the soil, enhancing carbon sequestration and reducing negative environmental impacts.

The exploitation of AFRs is, therefore, a relevant contribution to the clearing of stands and, consequently, the prevention of forest fires, and the impacts on soil fertility must be considered within the framework of responsible and sustainable management, obviously using good forestry practices. In fact, Portugal has implemented environmental policies to encourage the development of projects to provide a chain response to forest and agricultural leftovers, namely, the development of RBCCs and forwarding biomass to composting or energetic valorisation. These projects may be financed by the *Fundo Ambiental*, according to Notice n.º 18404/2023 [48]. Further efforts must be performed to optimise the management of AFRs and promote their valorisation and the use of organic matter in soils. Information should also be provided to landowners, in articulation with local entities, to promote awareness-raising, communication, and training actions on the importance of treating and valuing forest and agricultural leftovers.

## 5. Conclusions

Studying the environmental impacts of biomass valorisation is essential for identifying and implementing sustainable practices, thereby contributing to the overall goal of environmental, social, and economic sustainability. This study focused on the assessment of potential impacts associated with the mulching, composting, and energetic valorisation of AFRs. It was evaluated that the composting process is the one with the lowest overall impact in 9 out of 11 impact categories: producing compost from the biomass that is energetically utilised at the BPP is environmentally appropriate and plays an important role in the quality soil improvement and carbon sequestration. Mulching, on the other hand, has a very different environmental performance when compared with composting due to both the low nutrient content of the biomass used by the soil and the plants and the consequent release of a large amount of nutrient emissions. However, the energy valorisation of AFRs continues to be a valid option, mainly for its benefits in terms of the reduction in climate-changing gas emissions.

Along with the evaluation of the environmental impacts of the AFR management strategies, this work highlights the importance of the implementation of proper methods to accomplish AFR collection and storage with mechanised methods and good-quality equipment and tools. To achieve goals, policies, financial support, and awareness-raising, communication and training actions must be implemented.

This study is very important and useful for local decision makers whose role is to define policies to manage residual biomass arising from agricultural and forestry managing practices to support sustainable development.

**Supplementary Materials:** The following supporting information can be downloaded at: https://www.mdpi.com/article/10.3390/su16020630/s1, Figures S1 and S2 present the characteristics of the area under study. Table S1. Assessment of the residual biomass productivity rate. Table S2. Equivalent processes and materials in SimaPro and characterisation of the LCIA results. Tables S3–S5 show the category indicators results for mulching, composting, and energetic valorisation.

**Author Contributions:** I.B. conducted data analysis, performed funding acquisition, and conceptualised and revised the article. E.S. and J.F. were also evolved in the design of the experiment and reviewed and edited the article. R.R. collected data and performed the data analysis, wrote the original draft, along with R.S. and V.M. M.F. reviewed the article. All authors contributed to the preparation of the manuscript. All authors have read and agreed to the published version of the manuscript.

**Funding:** This research was funded by CISeD, grant number PI&Di/CISeD/003/2021.

**Institutional Review Board Statement:** Not applicable.

**Informed Consent Statement:** Not applicable.

**Data Availability Statement:** Data are contained within the article.

**Acknowledgments:** This work was financed by National Funds through FCT—Foundation for Science and Technology, I.P., under the project Ref.ᵃ UIDB/05583/2020. We additionally thank the Digital Services Research Center (CISeD) and the Polytechnic University of Viseu for the support provided. This work was elaborated within the scope of the project BioValor—Forest Ecopoint: Integrated Biomass Appreciation and Digitisation of its Management.

**Conflicts of Interest:** The authors declare no conflicts of interest.

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
