# Peer review of "Valorisation of Forest and Agriculture Residual Biomass—The Application of Life Cycle Assessment to Analyse Composting, Mulching, and Energetic Valorisation Strategies"

_sustainability, doi:10.3390/su16020630_

Round 1

Reviewer 1 Report

Comments and Suggestions for Authors

This paper conducts a comparative LCA study of three ways of valorisation of forest and agriculture residual biomass. The research problem of this article is interesting and valuable to knowledge. 

However, the authors stated this article is for decision-makers, the discussion section is more of an explanation of the results, the current manuscript does not present an efficient discussion on the empirical strategies, discussion from different perspectives and parameters, such as the land use, the ROI and so forth of different methods, together with the LCA results should be compared and further discussed. 

There are considerable assumptions in assessment data and comparative LCA, sensitive analysis is lacking for different scenarios.

In addition, there are various LCIA methods, and the choice of LCIA methods impacts the results and interpretation of this comparison, the reason for choosing the CML baseline method needs to be discussed.  

The manuscript is not completed, there are significant typos and size differences in the manuscript, references as well as format problems in supplementary materials.   

Comments on the Quality of English Language

Minor editing of English language required

Author Response

Thank you for your constructive feedback on our paper “Valorisation of forest and agriculture residual biomass – the application of life cycle assessment to analyze composting, mulching, and energetic valorisation strategies”. We appreciate the time and effort you've dedicated to reviewing our work and we think that the improvements done have effectively increased the article quality. Thank you so much.

Reviewer 2 Report

Comments and Suggestions for Authors

In the article, the researchers determined the importance of agricultural and forestry residues in harvesting residual biomass and converting it into biomass for electricity generation. Biomass valorization is the process of mulching and composting and the researchers determined the impact of utilization on the environmental impact sustainability index.

 Comments:

Abstract.

The abstract does not present a sufficient effect of the conducted research. The abstract lacks the provision of defined research results.

Introduction

The introduction initially presents the fire risk to the forests of Viseu, Portugal, as a result of the ignition of maintained and unused wood residues.

The introduction to the case study should only come later in the paper. The introduction should address the overall national and global impact of logging residues on both fire and biological (insect development) risks to forests.

Burning of forest and agricultural residues is certainly a serious problem especially for developing countries. What are the forest and agricultural residue abundance indicators for the study area of Europe? There was no verification of the quantitative or percentage indicator indicated in the literature.

The process of closed-loop circulation and valorization of agricultural and forestry residues has not been sufficiently presented in the literature. The scope of research in this area is very wide. The division of valorization methods is correctly presented.

Selected methods of valorization techniques through composting, mulching and energy use should find their place in the research methodology.

Methodology

The LCA of agricultural and forestry products is presented correctly referring to the standards and literature of the research subject. 

The methodological part includes a reference to the adopted areas of assessment of the processing cycle of agricultural and forestry residues, as mentioned in the introduction. The purpose and scope of the study were presented. The processes of processing agricultural and forestry residues were thoroughly characterized. What was missing was a reference to the literature describing the selected methods of processing evaluation.

Inventory data for LCA were presented. The dominance of pine as a forest species and eucalyptus in the coastal area was indicated. By-products (forest residues) are not defa who waste, but a recyclable product. General data characterizing the studied by-product were presented. A broader presentation of the properties and resources of the raw material was missing. The literature on the subject is very rich here.

The logistic cycle of residue movement and processing was described based on literature data. A measurement method was chosen, but it was not specified for which areas the method of movement of forest by-products is considered. The methodology shows how the productivity of post-agricultural by-products is calculated/ There was no specification of the area from which forest and post-agricultural residues were extracted.

The process of shredding, composting, mulching and energy valorization was described precisely with reference to complete mass data. Whether these results are the result of research or from specific sources is not stated. It would be reasonable to state the source of the data presented in the methodology.

The LCA impact assessment is presented comprehensively.

Results

The authors presented the results of LCA analyses for the use of forest and agricultural residues in the processes of composting , mulching and conversion to electricity.

A full identification of the cost of conversion to the environment was presented and the potential for the use of woody and agricultural biomass was considered. The role of combustion and composting waste management as ecosystem-supporting fertilizers is indicated.The total impact of mulching and composting was indicated.

Discussion

A coherent evaluation and discussion of the results obtained with reference to the literature was presented.

Unfortunately, there was no reference to life cycle modeling systems and closed product cycles for wood and post-agricultural residues. A broader reference to the literature on wood and post-agricultural biomass cycles was missing.

Conclusions

The important issue of researching the effect of mulching on a slight improvement in soil fertility quality has been raised. An important role is played by composting in securing agricultural and partially forestry productivity. The important direction of using biomass for energy purposes was rightly pointed out.  However, important indicators of valuing the environmental, social and economic impact in terms of energy security provided by stable biomass did not shine through in the conclusions.

Concretized summaries of the research results obtained from the discussion were missing.

 The paper is very interesting and brings new knowledge to the field of sustainable development of forest and agricultural biomass management.

Author Response

(The authors gave the same response as above.)

Reviewer 3 Report

Comments and Suggestions for Authors

This is an exceptionally thorough, clever, novel, and rigorously formulated work, with clear connections to sustainability as well as experiments.

Comments on the Quality of English Language

it is well written

Author Response

Thank you for your constructive feedback on our paper “Valorisation of forest and agriculture residual biomass – the application of life cycle assessment to analyze composting, mulching, and energetic valorisation strategies”. We appreciate the time and effort you've dedicated to reviewing our work. Nevertheless,  we think that the improvements done have effectively increased the article quality. Thank you so much.